# Identifiability of deep generative models without auxiliary information

**Bohdan Kivva**[*]
University of Chicago
bkivva@uchicago.edu

**Goutham Rajendran**[*]
University of Chicago
goutham@uchicago.edu

**Pradeep Ravikumar**
Carnegie Mellon University
pradeepr@cs.cmu.edu

**Bryon Aragam**
University of Chicago
bryon@chicagobooth.edu

## Abstract

We prove identifiability of a broad class of deep latent variable models that (a) have universal approximation capabilities and (b) are the decoders of variational autoencoders that are commonly used in practice. Unlike existing work, our analysis does not require weak supervision, auxiliary information, or conditioning in the latent space. Specifically, we show that for a broad class of generative (i.e. unsupervised) models with universal approximation capabilities, the side information $u$ is not necessary: We prove identifiability of the entire generative model where we do not observe $u$ and only observe the data $x$. The models we consider match autoencoder architectures used in practice that leverage mixture priors in the latent space and ReLU/leaky-ReLU activations in the encoder, such as VaDE and MFC-VAE. Our main result is an identifiability hierarchy that significantly generalizes previous work and exposes how different assumptions lead to different "strengths" of identifiability, and includes certain "vanilla" VAEs with isotropic Gaussian priors as a special case. For example, our weakest result establishes (unsupervised) identifiability up to an affine transformation, and thus partially resolves an open problem regarding model identifiability raised in prior work. These theoretical results are augmented with experiments on both simulated and real data.

## 1 Introduction

One of the key paradigm shifts in machine learning (ML) over the past decade has been the transition from handcrafted features to automated, data-driven representation learning, typically via deep neural networks. One complication of automating this step in the ML pipeline is that it is difficult to provide guarantees on what features will (or won't) be learned. As these methods are being used in high stakes settings such as medicine, health care, law, and finance where accountability and transparency are not just desirable but often legally required, it has become necessary to place representation learning on a rigourous scientific footing. In order to do this, it is crucial to be able to discuss ideal, target features and the underlying representations that define these features. As a result, the ML literature has begun to move beyond consideration solely of downstream tasks (e.g. classification, prediction, sampling, etc.) in order to better understand the structural foundations of deep models.

Deep generative models (DGMs) such as variational autoencoders (VAEs) [37, 56] are a prominent example of such a model, and are a powerful tool for unsupervised learning of latent representations, useful for a variety of downstream tasks such as sampling, prediction, classification, and clustering.

---

[*]Equal contribution

36th Conference on Neural Information Processing Systems (NeurIPS 2022).

Despite these successes, training DGMs is an intricate task: They are susceptible to posterior collapse and poor local minima [13, 24, 69, 72], and characterizing their latent space remains a difficult problem [e.g. 40, 67]. For example, does the latent space represent semantically meaningful or practically useful features? Are the learned representations stable, or are they simply artifacts of peculiar choices of hyperparameters? These questions have been the subject of numerous studies in recent years [e.g. 7, 12, 42, 45, 48, 60], and in order to better understand the behaviour of these models and address these questions, the machine learning literature has recently turned its attention to fundamental identifiability questions [14, 35, 69]. Identifiability is a crucial primitive in machine learning tasks that is useful for probing stability, consistency, and robustness. Without identifiability, the output of a model can be unstable and unreliable, in the sense that retraining under small perturbations of the data and/or hyperparameters may result in wildly different models.[1] In the context of deep generative models, the model output of interest is the latent space and the associated representations induced by the model.

In this paper, we revisit the identifiability problem in deep latent variable models and prove a surprising new result: Identifiability is possible under commonly adopted assumptions *and* without conditioning in the latent space, or equivalently, without weak supervision or side information in the form of auxiliary variables. This contrasts a recent line of work that has established fundamental new results regarding the identifiability of VAEs that requires conditioning on an auxiliary variable $u$ that renders each latent dimension conditionally independent [35]. While this result has been generalized and relaxed in several directions [10, 22, 23, 36, 39, 43, 50, 61, 73], fundamentally these results still crucially rely on the side information $u$. We show that this is in fact unnecessary—confirming existing empirical studies [e.g 18, 70]—and do so without sacrificing any representational capacity. What's more, the model we analyze is closely related to deep architectures that have been widely used in practice [16, 18, 32, 33, 41, 41, 44, 71]: We show that there is good reason for this, and provide new insight into the properties of these models and support for their continued use.

**Overview** More specifically, we consider the following generative model for observations $x$:

$$x = f(z) + \varepsilon, \quad x = (x_1, \ldots, x_n) \in \mathbb{R}^n, \quad z = (z_1, \ldots, z_m) \in \mathbb{R}^m, \tag{1}$$

where the latent variable $z$ follows a Gaussian mixture model (GMM),[2] $f : \mathbb{R}^m \to \mathbb{R}^n$ is a piecewise affine nonlinearity such as a ReLU network, and $\varepsilon \in \mathbb{R}^n$ is independent, random noise.[3] We do not assume that the number of mixture components, nor the architecture of the ReLU network, are known in advance, nor do we assume that $z$ has independent components. Both the mixture model and neural network may be arbitrarily complex, and we allow for the discrete hidden state that generates the latent mixture prior to be high-dimensional and dependent. This includes both vanilla VAEs (i.e. with a standard isotropic Gaussian prior) and classical ICA models (i.e. for which the latent variables are mutually independent) as special cases. Since both $z$ and $f$ are allowed to be arbitrarily complex, the model (1) has universal approximation capabilities, which is crucial for modern applications.

This model has been widely studied in the literature from a variety of different perspectives:

- *Nonlinear ICA.* When the $z_i$ are mutually independent, (1) recovers the standard nonlinear ICA model that has been extensively studied in the literature [1, 26–29, 74]. Although our most general results do not make independence assumptions, our results cover nonlinear ICA as a special case (see Section A.2 for more discussion).

- *VAE with mixture priors.* When the prior over $z$ is a mixture model (e.g. such as a GMM), the model (1) is closely related to popular autoencoder architectures such as VaDE [32], SVAE [33], GMVAE [16], DLGMM [52], VampPrior [66], MFC-VAE [18], etc. Although such VAEs with mixture priors have been used extensively in applications, theoretical results are missing.

- *Warped mixtures.* Another closely related model is the warped mixture model of Iwata et al. [31], which is a Bayesian version of (1). Once again, theoretical guarantees for these models are lacking.

---

[1]Formally, identifiability means the parametrization of the model is injective. See Section 2 for details.
[2]See Remark 2.1 for extensions to more general mixture priors.
[3]Our results include the noiseless case $\varepsilon = 0$ as a special case.

| Assumptions on $f$ | Assumptions on $Z$ | Theoretical guarantees | Result |
|:---:|:---:|:---:|:---:|
| (P1) | (F1), (F2) | $\mathbb{P}(Z)$ identifiable up to an affine transformation | Theorems 3.9(a), 3.10(a) |
| (P1) | (F1), (F4) | $\mathbb{P}(Z)$ and $f$ up to identifiable an affine transformation | Theorems 3.9(c), 3.10(d) |
| (P1), (P2) | (F1), (F4) | $\mathbb{P}(Z)$ and $f$ identifiable up to permutation, scaling and translation | Theorems 3.9(b), 3.10(b) |
| (P1), (P2), (P3) | (F1), (F4) | $\mathbb{P}(U, Z)$ and $f$ are identifiable up to permutation, scaling and translation | Theorems 3.10(c), 3.10(d) |

Table 1: Summary of results in this paper. The strength of the assumptions increases in each successive row, as do the strength of the guarantees. See Section 3.3 for formal statements.

- *iVAE.* Finally, (1) is also the basis of the iVAE model introduced by Khemakhem et al. [35], where identifiability (up to certain equivalences) is proved when there is an additional auxiliary variable $u$ that is observed such that $z_i \perp\!\!\!\perp z_j \,|\, u$.

**Contributions**    Driven by this recent interest from both applied and theoretical perspectives, our main results (Theorems 3.9, 3.10) show that the model (1) is identifiable up to various *linear* equivalences, without conditioning or auxiliary information in the latent space. In fact, we develop a hierarchy of results under progressively stronger assumptions on the model, beginning with affine equivalence and ending up with a much stronger equivalence up to permutations only. See Table 1 for a summary.

In order to develop this hierarchy, we prove several technical results of independent interest:

1. First, we establish a novel identifiability result for nonparametric mixtures (Theorem C.2);

2. Second, we show how to use the mixture prior to strengthen existing identifiability results for nonlinear ICA (Theorem D.1);

3. Third, we extend existing results [38] on the recovery of structured multivariate discrete latent variable models to recovery under an unknown affine transformation (Theorem F.2).

Our proof techniques—based on elementary tools from analytic function theory and mixture identifiability—are new and depart from existing work in this area. As a consequence, the analysis itself provides new insight into the structure and behaviour of deep generative models.

**Related work**    This problem is widely studied, and has garnered significant recent interest, so we focus only on the most closely related work here.

Classical results on nonlinear ICA [28] establish the nonidentifiability of the general model (i.e. without restrictions on $z$ and $f$); see also Darmois [15], Jutten et al. [34]. More recently, Khemakhem et al. [35] proved a major breakthrough by showing that given side information $u$, identifiability of the entire generative model is possible up to certain (nonlinear) equivalences. Since this pathbreaking work, many generalizations have been proposed [10, 22, 23, 36, 39, 43, 50, 61, 73], all of which require some form of auxiliary information. Other approaches to identifiability include various forms of weak supervision such as contrastive learning [75], group-based disentanglement [46], and independent mechanisms [20]. Non-identifiability has also been singled out as a contributing factor to practical issues such as posterior collapse in VAEs [69, 72].

Our approach is to avoid additional forms of supervision altogether, and enforce identifiability in a purely unsupervised fashion. Recent work along these lines includes Wang et al. [69], who propose to use Brenier maps and input convex neural networks, and Moran et al. [51] who leverage sparsity and an anchor feature assumption. Aside from different assumptions, the main difference between this line of work and our work is that their work only identifies the latent space $P(Z)$, whereas our focus is on jointly identifying *both* $P(Z)$ and $f$. In fact, we provide a decoupled set of assumptions

that allow $f$ or $P(Z)$ or both to be identified. Thus, we partially resolve in the affirmative an open problem regarding model identifiability raised by the authors in their discussion.

Another distinction between this line of work and the current work is our focus on architectures and modeling assumptions that are *standard* in the deep generative modeling literature, specifically ReLU nonlinearities and mixture priors. As noted above, there is a recent tradition of training variational autoencoders with mixture priors [16, 18, 32, 33, 41, 41, 44, 71]. Our work builds upon this empirical literature, showing that there is good reason to study such models: Not only have they been shown to be more effective compared to vanilla VAEs, we show that they have appealing theoretical properties as well. In fact, recent work [18, 70] has observed precisely the identifiability phenomena studied in our paper, however, this work lacks rigorous theoretical results to explain these observations.

Another related line of work studies identification in graphical models with latent variables, albeit without any explicit connection to deep generative models [17, 38, 49, 54].

Finally, since a key step in our proof involves the analysis of a nonparametric mixture model (see Appendix C for details), it is worth reviewing previous work in mixture models. See Allman et al. [2] for an overview. Of particular use for the present work are Teicher [63] and Barndorff-Nielsen [8], wherein the identifiability of Gaussian and exponential family mixtures, respectively, are proved. Specifically for nonparametric mixtures, existing results consider product mixtures [21, 64], grouped observations [57, 68], symmetric measures [9, 25], and separation conditions [4]. For context, we note here that a discrete VAE can be interpreted as a mixture model in disguise: This is a perspective that we leverage in our proofs. We are not aware of previous work in the deep generative modeling literature that exploits this connection to prove identifiability results.

## 2  Preliminaries

We first introduce the main generative model that we study and its properties, and then proceed with a brief review of identifiability in deep generative models.

**Generative model**   The observations $x \in \mathbb{R}^n$ are realizations of a random vector $X$, and are generated according to the generative model (1), where $z \in \mathbb{R}^m$ represents realizations of an unobserved random vector $Z$. We make the following assumptions on $Z$ and $f$:[4]

- (P1) $P(Z)$ is a (possibly degenerate) Gaussian mixture model with an unknown number of components $J \geq 1$, i.e.

$$p(z) = \sum_{j=1}^{J} \lambda_j \varphi(z; \mu_j, \Sigma_j), \quad \sum_{j=1}^{J} \lambda_j = 1, \quad \lambda_j > 0, \tag{2}$$

    where $p(z)$ is the density of $P(Z)$ with respect to some base measure, and $\varphi(z; \mu_j, \Sigma_j)$ is the gaussian density with mean $\mu_j$ and covariance $\Sigma_j$.

- (F1) $f$ is a piecewise affine function, such as a multilayer perceptron with ReLU (or leaky ReLU) activations.

Recall that an affine function is a function $x \mapsto Ax + b$ for some matrix $A$. As already discussed, special cases of this model have been extensively studied in both applications and theory, and both (P1)-(F1) are quite standard in the literature on deep generative models and represent a useful model that is widely used in practice [e.g. 16, 18, 32, 33, 41, 41, 44, 71]. In particular, when $J = 1$ this is simply a classical VAE with an isotropic Gaussian prior (see Section A.2 for more discussion).

*Remark* 2.1. The assumption that $P(Z)$ is a GMM can be replaced with more general exponential family mixtures [8] as long as (a) the resulting mixture prior $p(z)$ is an analytic function and (b) the exponential family is closed under affine transformations.

**Universal approximation**   Under assumptions (P1)-(F1), the model (1) has universal approximation capabilities. In fact, any distribution can be approximated by a mixture model (2) with sufficiently

---

[4]In the sequel, we will use (P#) to index assumptions on the prior $P(Z)$, and (F#) to index assumptions on the decoder $f$.

many components $J$ [e.g. 53]. Alternatively, when $J$ is bounded, by taking $f$ to be a sufficiently deep and/or wide ReLU network, any distribution can be approximated by $f(Z)$ [e.g. 47, 65], even if $f$ is invertible [30]. Thus, there is no loss in representational capacity in (P1)-(F1). To the best of our knowledge, our results are the first to establish identifiability of *both* the latent space and decoder for deep generative models *without* conditioning in the latent space or weak supervision. We note that Wang et al. [69] and Moran et al. [51] also propose deep architectures that identify the latent space, but not the decoder.

**Identifiability** A statistical model is specified by a (possibly infinite-dimensional, as in our setting) parameter space $\Theta$, a family of distributions $\mathcal{P}$, and a mapping $\pi : \Theta \to \mathcal{P}$; i.e. $\pi(\theta) \in \mathcal{P}$ for each $\theta \in \Theta$. In more conventional notation, we define $\mathcal{P} = \{p_\theta : \theta \in \Theta\}$, in which case $p_\theta = \pi(\theta)$. A statistical model is called *identifiable* if the parameter mapping $\pi$ is one-to-one (injective). In practical applications, the strict definition of identifiability is too strong, and relaxed notions of identifiability are sufficient. Classical examples include identifiability up to permutation, re-scaling, or orthogonal transformation. More generally, a statistical model is *identifiable up to an equivalence relation* $\sim$ defined on $\Theta$ if $\pi(\theta) = \pi(\theta') \implies \theta \sim \theta'$. For more details on the different notions of identifiability in deep generative models, see [35, 36, 58].

More precisely, we use the following definition. Let $f_\sharp P$ denote the pushforward measure of $P$ by $f$.

**Definition 2.2.** Let $\mathcal{P}$ be a family of probability distributions on $\mathbb{R}^m$ and $\mathcal{F}$ be a family of functions $f : \mathbb{R}^m \to \mathbb{R}^n$.

1. For $(P, f) \in \mathcal{P} \times \mathcal{F}$ we say that the prior $P$ is *identifiable (from $f_\sharp P$) up to an affine transformation* if for any $(P', f') \in \mathcal{P} \times \mathcal{F}$ such that $f_\sharp P \equiv f'_\sharp P'$ there exists an invertible affine map $h : \mathbb{R}^m \to \mathbb{R}^m$ such that $P' = h_\sharp P$ (i.e., $P'$ is the pushforward measure of $P$ by $h$).

2. For $(P, f) \in \mathcal{P} \times \mathcal{F}$ we say that the pair $(P, f)$ is *identifiable (from $f_\sharp P$) up to an affine transformation* if for any $(P', f') \in \mathcal{P} \times \mathcal{F}$ such that $f_\sharp P \equiv f'_\sharp P'$ there exists an invertible affine map $h : \mathbb{R}^m \to \mathbb{R}^m$ such that $f' = f \circ h^{-1}$ and $P' = h_\sharp P$.

If the noise $\varepsilon$ has a known distribution, then $f_\sharp P$ is identifiable from the convolution $(f_\sharp P) * \varepsilon$. Hence, this definition can be automatically extended to the setup with known noise. This definition also can be extended to transformations besides affine transformations (e.g. permutations, translations, etc.) in the obvious way.

Identifiability is a crucial property for a statistical model: Without identifiability, different training runs may lead to very different parameters, making training unpredictable and replication difficult. The failure of identifiability, also known as *underspecification* and *ill-posedness*, has recently been flagged in the ML literature as a root cause of many failure modes that arise in practice [14, 69, 72]. As a result, there has been a growing emphasis on identification in the deep learning literature, which motivates the current work. Finally, in addition to these reproducibility and interpretability concerns, identifiability is a key component in many applications of latent variable models including causal representation learning [59], independent component analysis [11], and topic modeling [3, 5]. See Ran and Hu [55] for additional discussion and examples.

**Auxiliary information and iVAE** It is well-known that assuming independence of the latent factors—i.e. $Z_i \perp\!\!\!\perp Z_j$—is insufficient for identifiability [28]. Recent work, starting with iVAE, shows identifiability by additionally assuming that a $k$-dimensional auxiliary variable $u$ is observed such that $p(z \mid u)$ is conditionally factorial, i.e. $Z_i \perp\!\!\!\perp Z_j \mid U$. This extra information serves to break symmetries in the latent space and is crucial to existing proofs of identifiability.

To make the connection with this work clear, observe that assumption (P1) is equivalent to assuming that there is an additional hidden state $U \in \{1, \dots, J\}$ such that $P(Z = z \mid U = j) = p_j(z)$ and $P(U = j) = \lambda_j$. More generally, $U = (U_1, \dots, U_k)$ may be multivariate. In this way, a direct parallel between our work and previous work is evident, with several crucial caveats:

- We do *not* assume that $U$ is observed—even partially—or known in any way;
- We allow for the $Z_i$ to be arbtrarily dependent even after conditioning on $U$, and this dependence need not be known;

- We do not even require the number of states $J$ to be known, and we do not require any bounds on $J$ (e.g. iVAE requires $J \geq m + 1$).

- In the case where $U$ is multivariate (i.e $k := \dim(U) > 1$), we do not require the number of latent dimensions $k$, the state spaces, or their dependencies to be known.

- The original iVAE paper only proves identifiability of $f$ up to a nonlinear transformation (see Lemma G.2 in Appendix G for details). By contrast, we will show identifiability of $f$ up to an affine transformation, without knowing $U$.

In order to break the symmetry without knowing anything about $U$ or its dependencies, we develop fundamentally new insights into nonparametric identifiability of latent variable models.

## 3 Main results

For any positive integer $d$, let $[d] = \{1, \ldots, d\}$. By (P1), we can write the model (1) as follows. Let $U = (U_1, \ldots, U_k) \in [d_1] \times \cdots [d_k]$ where $d_i := \dim(U_i)$ and $k := \dim(U)$; we allow $U$ to be multivariate ($k > 1$) and dependent—i.e., we do not assume that the $U_i$ are marginally independent. It follows trivially from (P1) that $P(U_1 = u_1, \ldots, U_k = u_k) \in \{\lambda_1, \ldots, \lambda_J\}$ and $J = \prod_i d_i$, where we recall that $J$ is the *unknown* number of mixture components in $P(Z)$. Denote the marginal distribution of $U$, which depends on $\lambda_j$, by $P_\lambda$. The variables $(U, Z)$ are unobserved and encode the underlying latent structure:

$$
\left.
\begin{aligned}
U = u &\sim P_\lambda(U = u) \\
[Z \,|\, U = u] &\sim N(\mu_u, \Sigma_u) \\
[X \,|\, Z = z] &\sim f(z) + \varepsilon, \quad \varepsilon \sim \mathcal{N}(0, \sigma^2)
\end{aligned}
\right\} \implies U \to Z \to X. \tag{3}
$$

Here, $P_\lambda$ is the distribution on $U$ described above. Our goal is to identify the latent distribution $P(U, Z)$ and/or the nonlinear decoder $f$ from the marginal distribution $P(X)$ induced by (3). We will additionally assume throughout that $m \leq n$; see Remark 3.6 for a discussion of the overcomplete case with $m > n$.

Our main results (Theorems 3.9-3.10) provide a hierarchy of progressively stronger conditions under which $P(U, Z)$, $f$, or both, can be identified in progressively stronger ways. The idea is to illustrate explicitly what conditions are sufficient to identify the latent structure up to affine equivalence (the weakest notion of identifiability we consider), equivalence up to permutation, scaling, and translation, and permutation equivalence (the strongest notion of identifiability we consider, and the strongest possible for any latent variable model).

We defer the statement of the main results to Section 3.3, after the main conditions have been described. As a preview to the main results, we first present the following corollary:

**Corollary 3.1.** *Suppose $k = \dim(U) = 1$, $J \geq 1$, $(U, Z)$ are unobserved, and $X$ is observed. (a) If $f$ is an invertible ReLU network, then both $P(U, Z)$ and $f$ are identifiable up to an affine transformation. (b) If $f$ is only weakly injective (cf. (F2)), then $P(U, Z)$ is still identifiable up to an affine transformation.*

For comparison, Corollary 3.1 already strengthens existing results, since $U$ is not required to be known and we are able to identify $f$. In fact, the latter answers an open question raised by Wang et al. [69]. What's more, this is just the *weakest* result implied by our main results: Under stronger assumptions on the latent structure, the affine equivalence presented above can be strengthened further.

Taken together, the results in this section have the following concrete implication for practitioners: For stably training variational autoencoders, there is now compelling justification to work with a GMM prior and deep ReLU/Leaky-ReLU networks. As we saw above, this is commonly done in practice already.

### 3.1 Possible assumptions on $f$

To distinguish cases where $f$ is and is not identifiable, we require the following technical definition. Recall that for sets $A, B$, $f^{-1}(A) = \{x : f(x) \in A\}$ and $f(B) = \{f(x) : x \in B\}$.

**Definition 3.2.** Let $m \leq n$ (see Remark 3.6) and $f : \mathbb{R}^m \to \mathbb{R}^n$.

(F2) We say that $f$ is *weakly injective* if (i) there exists $x_0 \in \mathbb{R}^n$ and $\delta > 0$ s.t. $|f^{-1}(\{x\})| = 1$ for every $x \in B(x_0, \delta) \cap f(\mathbb{R}^m)$, and (ii) $\{x \in \mathbb{R}^n : |f^{-1}(\{x\})| = \infty\} \subseteq f(\mathbb{R}^m)$ has measure zero with respect to the Lebesgue measure on $f(\mathbb{R}^m)$.

(F3) We say that $f$ is *observably injective* if $\{x \in \mathbb{R}^n : |f^{-1}(\{x\})| > 1\} \subseteq f(\mathbb{R}^m)$ has measure zero with respect to the Lebesgue measure on $f(\mathbb{R}^m)$. In other words, $f$ is injective for almost every $x$ in its image $f(\mathbb{R}^m)$ (i.e. almost every "observable" $x$).

(F4) We say that $f$ is *injective* if $|f^{-1}(\{x\})| = 1$ for every $x \in f(\mathbb{R}^m)$.

*Remark* 3.3. For piecewise affine functions assumption (F2) is weaker than assumption (F3), which in turn is weaker than (F4). Therefore, for piecewise affine functions we have the chain of implications:

$$\text{(F4)} \implies \text{(F3)} \implies \text{(F2)}.$$

In the sequel, we mostly focus on (F2) and (F4) for simplicity; although we prove results for (F3) in Appendix D.1. See also Remarks 3.5, 3.11.

**Example 3.4.** In general, a deep ReLU network may be either injective or observably injective, or neither (e.g. $\text{ReLU}(-\text{ReLU}(x)) = 0$). For example, although $x \mapsto \text{ReLU}(x)$ is not injective, it is observably injective, where $\text{ReLU}(x) = \max\{0, x\}$ is the usual rectified linear unit. To see this, note that image of ReLU is the set $\mathbb{R}_{\geq} = \{y \mid y \geq 0\}$, and ReLU has the unique preimage for every $y \in \mathbb{R}_{>} = \{y \mid y > 0\}$. Clearly, $(\mathbb{R}_{\geq} \setminus \mathbb{R}_{>}) = \{0\}$ has measure zero inside $\mathbb{R}_{\geq}$.

At the same time, $x \mapsto 0$ and $x \mapsto |x|$ are not even weakly injective.

*Remark* 3.5. In Appendix H, we show that ReLU networks or Leaky ReLU networks are generically observably injective (and hence also weakly injective) under simple assumptions on their architecture.

*Remark* 3.6. We restrict attention to the case $m \leq n$, which is a standard assumption, as it is common to think of a latent space to be a low-dimensional representation of the observed space. In the overcomplete case, i.e. when $m > n$, we believe that identifiability is unlikely unless stronger assumptions are made, or weaker notions of identifiability are considered. To see this, consider the projection $f(x, y) = x$, which is trivially affine. Then we can arbitrarily transform the $y$-coordinate without changing $P$, i.e. $(f \circ g)_\sharp P = f_\sharp P$, where $g(x, y) = (x, h(y))$ for any $h$. As an example of identifiability in the overcomplete regime under stronger assumptions, when the auxiliary variable $u$ is known, [36] show that the feature maps $f$ and $g$ in conditional energy-based models (for which $p(x \mid u) \propto \exp(f(x)^T g(u))$) can be identified up to an affine transformation.

## 3.2 Possible assumptions on $Z$

Our weakest result requires no additional assumptions on $Z$ beyond (P1); see Corollary 3.1. Under stronger assumptions, more can be concluded. As with the previous section, the assumptions presented here are not necessary, but may be imposed in order to extract stronger results.

The first condition is a mild condition that allows us to strengthen affine identifiability:

(P2) $Z_i \perp\!\!\!\perp Z_j \mid U$ for all $i \neq j$ and there exist a pair of states $U = u_1$ and $U = u_2$ such that all $((\Sigma_{u_1})_{tt} / (\Sigma_{u_2})_{tt} \mid t \in [m])$ are distinct. (Note that this implies $J \geq 2$).

The second condition is more technical, and is only necessary if $k > 1$ and we wish to identify $P(U)$ in addition to $P(Z)$. In fact, not only will we recover $P(U)$, but also the (unknown) number of hidden variables (i.e. $k$) and their state spaces (i.e. $d_j$). Note that $P(U)$ is not needed to sample from (1), as long as we have $P(Z)$. Before introducing this condition, we need a preliminary definition.

**Definition 3.7.** Let $U_{-i}$ denote $\{U_j : j \neq i\}$. We define $\text{ne}(U_i) = [m] \setminus \{t : Z_t \perp\!\!\!\perp U_i \mid U_{-i}\}$ and $\text{ne}(Z_i) = \{t : Z_i \in \text{ne}(U_t)\}$. For a subset $Z' \subset Z$, $\text{ne}(Z') = \cup_{Z_i \in Z'} \text{ne}(Z_i)$.

The neighborhood $\text{ne}(U_i)$ collects the variables $Z_t$ that depend on $U_i$ directly.

(P3) The following conditions hold:

(a) For all $Z' \subset Z$ and $u_1 \neq u_2$, $P(Z' \mid \text{ne}(Z') = u_1) \neq P(Z' \mid \text{ne}(Z') = u_2)$;

(b) If $P(U', Z, X) = P(U, Z, X)$, then $\dim(U') \leq \dim(U)$; and

(c) For any $U_i \neq U_j$ the set $\mathrm{ne}(U_i)$ is not a subset of $\mathrm{ne}(U_j)$.

Condition (P3) is a "maximality" condition that is adapted from Kivva et al. [38]: We are interested in identifying the most complex latent structure with the most number of hidden variables. This is in fact necessary since we can always merge two (or more) hidden variables into a single hidden variable without changing the joint distribution. Moreover, if two distinct hidden variables $U_i \neq U_j$ have the same neighborhood (or one is a subset of another), then it is known that $P(U)$ cannot be identified [17, 38, 54]. Evidently, if we seek to learn $P(U)$ in addition to $P(Z)$, then this must be avoided. Finally, as the proof will indicate, this condition is slightly stronger than what is needed (see Remark F.3 for details).

*Remark* 3.8. Condition (P3) should be contrasted with the stronger "anchor words" assumption that has appeared in prior work [5, 6, 51]: In fact, the existence of an anchor word for each $U_j$ automatically implies that $\mathrm{ne}(U_i)$ is not a subset of $\mathrm{ne}(U_j)$ for $i \neq j$. Thus, anchor words are a sufficient but not necessary condition for identifiability, whereas Condition (P3) is indeed necessary as described above.

More details and discussion on these assumptions can be found in Appendix F.

### 3.3 Main identifiability results

When $\dim(U) = 1$, there is no additional structure in $U$ to learn, and so the setting simplifies considerably. We begin with this special case before considering the case of general multivariate $U$.

**Theorem 3.9.** *Assume* $\dim(U) = 1$. *Under (P1)-(F1), we have the following:*

(a) *(F2)* $\implies$ *$P(U, Z)$ is identifiable from $P(X)$ up to an affine transformation of $Z$.*

(b) *(F2)+(P2)* $\implies$ *$P(U, Z)$ is identifiable from $P(X)$ up to permutation, scaling, and/or translation of $Z$.*

(c) *In either (a) or (b), if additionally (F4) holds and $f$ is continuous, then $f$ is also identifiable from $P(X)$ up to an affine transformation.*

The next result generalizes Theorem 3.9 to arbitrary (possibly multivariate) discrete $U$. This is an especially challenging case: Unlike previous work such as iVAE that assumes $U$ (and hence its structure) is known, we do not assume anything about $U$ is known. Thus, everything about $U$ must be reconstructed based on $P(X)$ alone, hence the need for (P3) to identify $P(U)$ below.

**Theorem 3.10.** *Under (P1)-(F1), we have the following:*

(a) *(F2)* $\implies$ *$P(Z)$ is identifiable from $P(X)$ up to an affine transformation.*

(b) *(F2)+(P2)* $\implies$ *$P(Z)$ is identifiable from $P(X)$ up to permutation, scaling, and/or translation.*

(c) *(F2)+(P2)+(P3)* $\implies$ *$(k, d_1, \ldots, d_k, P(U))$ are identifiable from $P(X)$ up to a permutation of $U$, and $P(Z)$ is identifiable up to permutation, scaling, and/or translation.*

(d) *In any of (a), (b), or (c), if additionally (F4) holds and $f$ is continuous, then $f$ is also identifiable from $P(X)$ up to an affine transformation.*

Without (P3), Kivva et al. [38] have shown that it is not possible to recover the high-dimensional latent state $U$, however, we can still identify the continuous latent state $Z$, which is enough to generate random samples from the model (1). In order to have fine-grained control over the individual variables in $U$, however, it is necessary to assume (P3).

*Remark* 3.11. If (F4) is relaxed to (F3) $f$ may not be identifiable up to an affine transformation, but it is "essentially" identifiable in the following sense. Let $S = \{x : |f^{-1}(\{x\})| > 1\}$. On every connected component of $\mathbb{R}^m \setminus f^{-1}(S)$, $f$ is identifiable up to an affine transformation (which may depend on the connected component). Note, for $f$ defined by a ReLU NN, points of $S$ are atoms of $P(X)$.

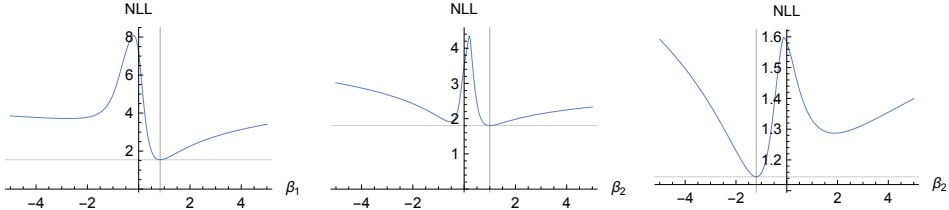

Figure 1: Selected examples of the negative log-likelihood for different runs. In each figure, one parameter from a model (e.g. $\beta_j$ is a weight in the neural network defining $f$) is selected, and the value of the negative log-likelihood is visualized as a function of this parameter. Vertical lines indicate the ground truth and (global) minimizer, which always coincide. Three particularly interesting, nonconvex examples are shown here. See Appendix J.3.1 for details.

*Remark* 3.12. If the assumption (F2) that $f$ is weakly injective is removed, then the claim of Theorem 3.9 is not true anymore. Consider $g(x) = f(x) = |x|$ and

$$
\begin{aligned}
P &= \frac{1}{3}N(-2, \sigma^2) + \frac{1}{3}N(-1, \sigma^2) + \frac{1}{3}N(3, \sigma^2) \quad \text{and} \\
P' &= \frac{1}{3}N(-2, \sigma^2) + \frac{1}{3}N(1, \sigma^2) + \frac{1}{3}N(3, \sigma^2).
\end{aligned}
\tag{4}
$$

It is easy to verify that $P$ cannot be transformed into $P'$ by an affine transformation, but $f_\sharp P$ and $g_\sharp P'$ are equally distributed.

*Remark* 3.13. In Theorems 3.9(a) and 3.10(a), the identifiability up to an affine transformation is the best possible if no additional assumptions on $Z$ are made (i.e. beyond (P1)). Indeed, for an arbitrary invertible affine map $h : \mathbb{R}^m \to \mathbb{R}^m$, $h(Z)$ has a GMM distribution, $f \circ h^{-1}$ is an invertible piecewise affine map, and $(U, Z, f)$ and $(U, h(Z), f \circ h^{-1})$ in model (3) generate the same distribution.

## 4 Experiments

There has been extensive work already to verify empirically that the model (1) under (P1)-(F1) is identifiable. For example, [70] observe that deep generative models with clustered latent spaces are empirically identifiable, and compared this directly to models that rely on side information, and [18] show that meaningful latent variables can be learned consistently in a fully unsupervised manner even when $U$ has high-dimensional structure. Moreover, [18] indicate that high-dimensional structure is important for improved performance. Beyond these, it is well-known that VAEs with mixture priors such as VaDE [32] achieve competitive performance on many benchmark tasks; see [16, 18, 33, 41, 41, 44, 71] for additional experiments and verification. Building upon the established success of these methods, we augment these experiments as follows: 1) We use simple examples to verify that the likelihood indeed has a unique minimizer at the ground truth parameters; 2) We train VaDE on (misspecified) simulated toy models; and 3) We measure stability (up to affine transformations) of the learnt latent spaces on real data. To measure this, we report the Mean Correlation Coefficient [36, Appendix A.2] metric, which is standard, and an $L^2$-based alignment metric (denoted by $\text{dist}_{\text{Aff}, L2}$). Definitions of these metrics and additional details on the experiments can be found in Appendix J.

**Maximum likelihood** We simulated models satisfying (P1)-(F1) by randomly choosing weights and biases for a single-layer ReLU network and randomly generating a GMM with $J = 2$ or 3 components. These models are simple enough that exact computation of the MLE along the likelihood surface is feasible via numerical integration (Figure 1). In all our simulations (50 total), the ground truth was the unique minimizer of the negative log-likelihood, as predicted by the theory. These examples also illustrate a small-scale test of misspecification in the theoretical model: We include cases where $J$ is misspecified and $f$ fails to satisfy (F4), but the MLE succeeds anyway.

**Simulated data** In our experiments on synthetic datasets we consider, to obtain an experimental evidence of identifiability of model (3) we fit VaDE to observed data 5 times (see Figure 2). Let

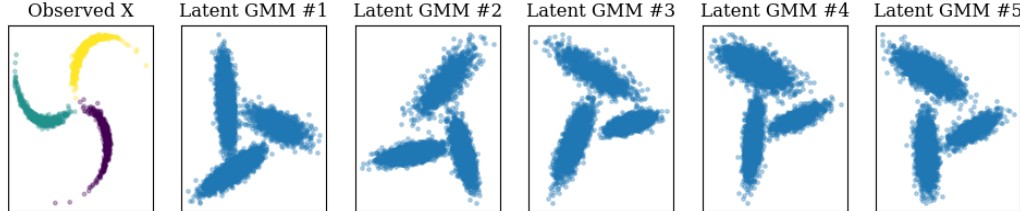

Figure 2: Recovered latent spaces for 5 runs of VaDE on pinwheel dataset with 3 clusters

$Z^{(1)}, Z^{(2)}, \ldots, Z^{(5)}$ be the learned latent spaces. For every pair $Z^{(i)}, Z^{(j)}$ we evaluate the MCC and $\mathrm{dist}_{\mathrm{Aff},L2}$ loss. For instance, for the pinwheel dataset with three clusters as in Figure 2, the average $\mathrm{dist}_{\mathrm{Aff},L2}(p_1, p_2)$ across 20 pairs $Z^{(i)}, Z^{(j)}$ is 0.113 with standard deviation 0.065. The average weak MCC is 0.87 and the average strong MCC is 1.0. This shows strong evidence of recovery of the latent space up to affine transformations.

**Real data**  We measure stability of the learnt latent space by training MFCVAE [18] on MNIST 10 times with different initializations and then comparing the latent representations learnt. It becomes computationally infeasible to compute $\mathrm{dist}_{\mathrm{Aff},L2}$ therefore we report only MCC. The strong MCCs are computed to be 0.7 (ReLU), 0.69 (LeakyReLU) and the weak MCCs are computed to be 0.91 (ReLU), 0.94 (LeakyReLU). These observations validate the observations first made in [70], who ran extensive experiments on VaDE and iVAE on several large datasets including MNIST, SVHN and CIFAR10. These strong correlations confirm our theory and are of particular importance to practitioners for whom stability of learning is of the essence.

## 5  Conclusion

We have proved a general series of results describing a hierarchy of identifiability for deep generative models that are currently used in practice. Our experiments confirm both on exact and approximate simulations that identifiability indeed holds in practice. An obvious direction for future work is to study finite-sample identifiability problems such as sample complexity and robustness (i.e. how many samples are needed to ensure that the global minimizer of the likelihood is reliably close to the ground truth?). Theoretical questions aside, developing a better understanding of the ELBO and its effect on optimization is an important practical question. For example, an important limitation of the current set of results is that they apply only to the likelihood, which is known to be nonconvex and intractable to optimize (see Figure 1 for concrete examples). It is an important open question to use these insights to develop better algorithms and optimization techniques that work on finite-samples with misspecified models (i.e. real data).

More generally, although our assumptions map onto architectures and priors that are widely used in practice, it is important to emphasize the relevant distinction between models and estimators. That is, the architectures used in practice represent the *estimators* used, and may not reflect realistic assumptions on the *model* itself (which is typically misspecified). For example, the piecewise affine assumption may not accurately reflect valid assumptions about real-world problems. Given the lack of purely unsupervised, nonparametric identifiability results in the literature, we view our results as an important technical step towards understanding practical identifiability for deep generative models. Thus, an important future direction is to replace our assumptions with more appropriate modeling assumptions that are relevant for practical applications.

## 6  Acknowledgements

We thank anonymous reviewers for useful comments and suggestions. G.R. was partially supported by NSF grants CCF-1816372 and CCF-200892. B.A. was supported by NSF IIS-1956330, NIH R01GM140467, and the Robert H. Topel Faculty Research Fund at the University of Chicago Booth School of Business. P.R. was supported by ONR via N000141812861, and NSF via IIS-1909816, IIS-1955532, IIS-2211907.

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
