# OpenReview forum: "Identifiability of deep generative models without auxiliary information"
_NeurIPS.cc/2022/Conference — NeurIPS 2022 Accept_

### Official Review · Reviewer_w4oC · 2022-07-11

**Rating:** 6
**Confidence:** 2
**Soundness:** 4 excellent
**Presentation:** 4 excellent
**Contribution:** 2 fair

**Summary:**

This paper explores the problem of DGMs' identifiability. The main contribution is theoretical as it proves a sequence of theorems about the identifiability of DGMs under different assumptions on decoder functions and latent distributions. It also has several experiments trying to verify the theorems on synthetic data and MNIST.

**Questions:**

1. The theorems in sec. 3 show that under certain conditions, all global minima of a DGM are the same up to an equivalence relation. But the current gradient based training of DGMs cannot guarantee finding a global minimum. Is there a way to analyze the identifiability of learned parameters of a DGM?

**Limitations:**

The authors have discussed limitations.

**Strengths And Weaknesses:**

I have some experience on deep generative models, but I'm not quite familiar with the problem of identifiability. As a result, my review might not be very targeted and discussions are very welcome from the authors and other reviews.

Pros:
1. Identifiability of DGMs is a very interesting question. It's an important path towards more reliable machine learning systems. It will also make analyzing DGMs easier.
2. This paper discusses several different assumptions for Z and f in detail, and proves how strong identifiability we can achieve under each assumption. The conclusion doesn't rely on knowing any auxiliary variable and provides a more comprehensive characterization of the problem.
3. The paper is well written. It starts from simple cases and gradually delves into details, which makes it a pleasure to read it.

Cons:
1. The theorem lacks a stability analysis. All theorems in this paper require the data distribution to exactly match. It's not clear what's the situation like if we allow a small mismatch between the real and learned data distributions. Taking Eqn. 4 in the paper as an example, we can find f and f', which maps P and P' to f#(P) and f'#(P'), which can be arbitrarily close to each other, but cannot exactly match because of the piecewise affine assumption.
2. The experiments fail to convince me that the theory can be applied to more complicated problems. The synthetic data is simply a transformation of three-component MoG. The real data is MNIST, which has obvious clustering property. Both datasets are too simple.

---

> ### Author Response · Authors · 2022-08-02
> **Response to Reviewer w4oC**
>
> We thank the reviewer for their careful reading of our work and their insightful questions. While we agree that these directions are of significant interest, we hasten to point out an important point: **Without identifiability, it is not possible to do a stability analysis or prove theorems about the consistency of a particular learning algorithm**. Indeed, this is why identifiability is so important, as it enables such practically important investigations. Thus, we view our work as a critical first step in this direction.
>
> We now address the comments and questions in more detail:
>
> > _Is there a way to analyze the identifiability of learned parameters of a DGM?_
>
> By “analyze the identifiability of the learned parameters”, we assume the reviewer means to prove there is a practically useful algorithm that successfully returns the true, identifiable parameters. (Please clarify if this is not what you mean.) This is a great question, and indeed ultimately what the community would like to develop. As outlined above, it is important to appreciate that our results are a *necessary* condition for such a result, and thus represent an important step in this direction. In other words, if the true parameters are *not* identifiable, then no learning algorithm can ever recover the true parameters consistently.
>
> Moreover, previous work has established that non-identifiability is closely related to failure modes of VAEs observed in practice, see e.g. the references at L180-188. One specific example is posterior collapse. See also L107-108, where we point out that non-identifiability leads to practical issues in deep generative modeling, such as posterior collapse in VAE training.
>
> On this line of thought, a closely related question that has been studied before is stability of learning algorithms, i.e. will repeated runs of the same algorithm produce the same parameters? This is also connected to the notion of algorithmic robustness. For VAEs, our experiments as well as the extensive experiments in prior works have empirically observed stability of training (see [1] and references therein). Connecting back to identifiability, we remark that in order to study stability of learning algorithms, it’s a necessary first step to study identifiability of the underlying parameters. Because if the underlying parameters are not identifiable, then repeated runs of an algorithm could possibly give different outputs yet being completely correct. In this regard, our work shows identifiability formally for VAE training under mixture priors.
>
> > _The theorem lacks a stability analysis. All theorems in this paper require the data distribution to exactly match._
>
> This is a great point. A stability analysis for our theory would indeed be highly desirable. However, this is highly nontrivial because neural networks can approximate any distribution to arbitrary accuracy. This situation is exacerbated by the fact that we only have access to finite samples in real life, see also L384-388. We leave this study for future work. (Furthermore, as we commented above, our results are nonetheless a necessary step towards answering questions about stability.)
>
> > _The experiments fail to convince me that the theory can be applied to more complicated problems._
>
> We highlight that extensive prior works (e.g. [1, 2, 3]) have already validated our results experimentally in a variety of settings, e.g. on MNIST, CIFAR10 and SVHN. Please see also L117-124 and L345-352 where we discuss this and provide additional (>8) citations. We emphasize that our experiments are meant to augment these prior works, which have already demonstrated practical aspects of this model. Therefore to translate our results into practice, both our experiments as well as these prior works should be taken together into consideration.
>
> **References:**
>
> [1] M. Willetts and B. Paige. I don’t need u: Identifiable non-linear ica without side information. 2021.
>
> [2] Z. Jiang, Y. Zheng, H. Tan, B. Tang, and H. Zhou. Variational deep embedding: An unsupervised and generative approach to clustering. 2016.
>
> [3] F. Falck, H. Zhang, M. Willetts, G. Nicholson, C. Yau, and C. C. Holmes. Multi-facet clustering variational autoencoders. 2021.

---

> > ### Comment · Reviewer_w4oC · 2022-08-09
> > **Acknowledgement**
> >
> > Thanks for the response! I will maintain my positive score.

---

### Official Review · Reviewer_SEs8 · 2022-07-13

**Rating:** 7
**Confidence:** 2
**Soundness:** 3 good
**Presentation:** 3 good
**Contribution:** 4 excellent

**Summary:**

This paper introduces new theoretical results for the identifiability of Variational Autoencoders with Gaussian mixture priors under several progressively stronger conditions on the latent distribution and the decoder (such as the decoder being a piecewise-affine function). The work extends and generalizes previous results on the identifiability of deep generative models, such as the iVAE. The key contribution is identifiability results that do not depend on auxiliary variables being observed. For example, the authors show that that under weak assumptions P(Z) is identifiable from P(X) up to an affine transformation and that additional the decoder parameters are also identifiable if the function family is injective.

The authors also include experiments on toy data and MNIST to verify the identifiability results empirically.

**Questions:**

I'm not sure I fully understand the definition of weak injectivity here (F2). What does $B(x_0, \delta)$ denote? Can you give a more intuitive explanation?

What is mean by "in the sequel"? Is that referring to the subsequent section?

**Limitations:**

The authors address some limitations and future work in the conclusion and discuss related negative results in the main text. Broader impacts are not discussed, though this is a largely theoretical work.

**Strengths And Weaknesses:**

The primary strength of this work is that it shows that modeling choices already commonly made in practice for Variational Autoencoders, such as ReLU networks and Gaussian mixture priors, maintain identifiability (up to affine transforms) for the latent codes and possibly the decoder parameters. This strengthens previous work, and makes a useful contribution to the theoretical analysis of deep generative models. It also helps to justify modeling practices which are commonly used, even in the more common case where auxiliary information is not available. In my opinion, the main results presented in theorems 3.9 and 3.10 are a sufficiently novel and useful contribution to the field. (I have not yet had a chance to fully verify the correctness of the proofs provided for these results, the supplement provides extensive details on the theoretical results).

The paper also generalizes to more complex cases with multiple discrete latent groups and more general exponential family mixtures.

The paper is also well-written, generally providing the necessary background, the key contributions, the assumptions and results in a clear, well-organized manner, with a few exceptions, see below.

The experiments section could use improvement, though understandably space for this section is limited. It is not quite clear to me what figure 1 results are showing. The likelihood as a function of decoder parameters? Latent codes? The right and middle plots have the same x-axis label. Figure 2 is somewhat more clear.

---

> ### Author Response · Authors · 2022-08-02
> **Response to Reviewer SEs8**
>
> We thank the reviewer for their review and positive feedback. Below we respond to the main concerns and are happy to discuss further as needed.
>
> > _I'm not sure I fully understand the definition of weak injectivity here (F2). What does B(x_0, \delta) denote? Can you give a more intuitive explanation?_
>
> $B(x_0, \delta)$ is the ball around $x_0$ of radius $\delta$. Formally, $B(x_0, \delta) = {x : \Vert x - x_0\Vert \le \delta}$ where $\Vert\cdot\Vert$ is the L2-norm.
>
> > _What is mean by "in the sequel"? Is that referring to the subsequent section?_
>
> We apologize for the confusion, sequel simply refers to later in the paper. We will try to reduce our usage of this term.
>
> > _It is not quite clear to me what figure 1 results are showing. The likelihood as a function of decoder parameters? Latent codes? The right and middle plots have the same x-axis label._
>
> We apologize for the confusion. With the extra page in the camera ready, we will add more descriptive details for this figure. Figure 1 shows how the negative log-likelihood varies as a function of the x-axis parameter (please see App J.3.1 for details on the precise definitions of the neural network parameters $\beta_j$)—since this is a high-dimensional function, we can only show certain projections for visualization purposes. The point is to show that indeed the ground truth is also the minimizer of the negative log-likelihood.
>
> Due to space limitations, we have presented three selected examples from different runs that are particularly interesting (e.g. due to nonconvexity and multiple local minima). The middle and right figures are showing the behavior of the negative log-likelihood wrt $\beta_2$ for two *different* simulated models. We will make this more clear in the camera ready.

---

### Official Review · Reviewer_dpBS · 2022-07-24

**Rating:** 8
**Confidence:** 3
**Soundness:** 4 excellent
**Presentation:** 3 good
**Contribution:** 4 excellent

**Summary:**

This paper proves a series of identifiability results for commonly used modern generative models composed of Gaussian mixture priors and piecewise affine (ReLU) decoder networks. Up to affine or simpler transformations on the latent space for the decoder function, the latent space and/or the decoder function can be specified given a learned distribution. Unlike previous studies on more generic network types, this framework does not require additional auxiliary information. The assumptions made for the prior and decoder are more restrictive but correspond to common practice. As far as I understand, the main intuition for the proofs come from identifiability of Gaussian mixture models (in reduced form) and nice but non-obvious properties of piecewise-affine functions. For example, each affine segment transforms a GMM into another GMM, which is identifiable.

**Questions:**

## Detailed suggestions.
1. Line 176, I think here the $f'$ and $P'$ inherit from the previous point, so perhaps should clarify that. Only $f$ is shared with the definition in line 172.
2. Line 181, the paper's focus is mostly on the functions rather than parameters, so should perhaps change "parameters" to "functions".
3. Line 183. "Failure model" may be understood as a (likelihood) performance failure, interpretability failure or identifiability failure. I would try to specify a little.
2. Definition 3.7. does $ne(U_i)$ refer to the set of indices or variables? The definition suggests it's an index set, but later uses assume they are variables.
2. Line 331, the $x$ inside $f^{-1}$ should be inside braces.
3. Line 812, should $h$ be a better function symbol to use as an example?
3. Definition C.8, there is a condition that the preimage is finite, but then in 859 the subspaces covering the non-affine supports can be infinite. I think the authors probably mean "countable" in Definition C.8, so that the preimage are made of atoms rather than continuous regions.
2. Throughout the development of the proofs, $x$ can appear as the input or output of the decoder function $f$, which can be quite confusing and takes time for the reader to follow.
2. equation (11), I think the notation $p|_{\cdot}$ deserves a specific definition.
2. Various symbols on page 20 can be much better illustrated by an example of $f\in\mathbb{R}\to\mathbb{R}$ that is piecewise affine, where $x_0$ is a value on the vertical axis, then y's on the horizontal axis, B is a region around one $y_i$, etc. The concept here is actually quite simple, though notationally heavy, for a reader like me.

**Limitations:**

I don't see any concerning limitations of the analyses.

**Strengths And Weaknesses:**

## Strengths:
1. To my knowledge, this paper is the first to set out rigorously the conditions needed for identifiability in deep generative models without auxiliary labels. Although the assumptions made are stronger than in previous work, they correspond to standard practice and thus are of great significance for the field.
1. The authors present a series of analyses from weaker to stronger assumptions of the model, demonstrating a substantial contribution to the field.
2. Writing is another strength of the paper. The main text gave a comprehensive overview of previous work and discussed the subtle details of the assumptions and concepts. Some crucial examples are given to clarify the definitions. These are presented in a well-designed layout that I find things very clear and easy to refer back and forth.
2. The appendix has a lot of background information and analysis compared to very relevant literature, including the iVAE.
2. Due to limited time availability, I checked the proof up to page 23 of the appendix. They seem correct to me and make intuitive sense, but I may be wrong for very technical details.

##  Weakness:
1. The main text, among a very clear introduction of the background and problem setup, is a repetitive sale of the achievement compared to the existing literature rather than a substantive and intuitive description of the theory. I'd encourage the authors to try to reduce the significance statement of the work, however impactful it may be. For example, the fact that "the theory does not assume any additional information" appears so many times in the main text that some readers may find it annoying.
2. The main numerical results should be listed in a small table or figure, some results currently in the appendix should also appear in the main text.

---

> ### Author Response · Authors · 2022-08-02
> **Response to Reviewer dpBS**
>
> We thank the reviewer for their detailed reading of the paper and their positive feedback. We are glad that the reviewer liked our writing clarity and deemed our contributions substantial. We appreciate the numerous suggestions for improvement and will incorporate these changes into the camera ready, including revising (/reducing) the language regarding significance and the presentation of the numerical results.
>
> Below, we address your questions:
>
> > _1. Line 176, I think here the f’ and P’ inherit from the previous point, so perhaps should clarify that. Only f is shared with the definition in line 172._
>
> Thank you, yes, they are notations from the previous point. We will clarify this.
>
> > _2. Line 181, the paper's focus is mostly on the functions rather than parameters, so should perhaps change "parameters" to "functions"._
>
> Thank you for flagging this as it is potentially confusing: Perhaps “outputs” is a more appropriate choice of word. In fact, our paper is interested in both: the function as well as the parameters of the latent probability distribution. Moreover, in applications of interest, f is a neural network, which is also defined by parameters. So, “outputs” might help to prevent confusion, as this is intended to be a generic discussion concerning general statistical models.
>
> > _3. Line 183. "Failure model" may be understood as a (likelihood) performance failure, interpretability failure or identifiability failure. I would try to specify a little._
>
> We are happy to clarify this further in the camera ready. The failure modes we are referring to are posterior collapse and non-identifiability, as discussed in the cited references.
>
> > _4. Definition 3.7. does ne(U_i) refer to the set of indices or variables? The definition suggests it's an index set, but later uses assume they are variables._
>
> Thank you for pointing this out. ne(U_i) by definition is an index set (which has a trivial bijection with a set of variables). In the proofs in the appendix, we indeed used it interchangeably for both indices and the corresponding variables. This is a standard convention in some (but not all) parts of the literature, so we will be sure to add a remark to clarify this point.
>
> > _5. Line 331, the x inside f^(-1) should be inside braces._
>
> Thank you, corrected.
>
> > _6. Line 812, should h be a better function symbol to use as an example?_
>
> In line 812, we think of f as of a function from Eq (1) (a function from L806). In the text, we denote them by f or g.
>
> > _7. Definition C.8, there is a condition that the preimage is finite, but then in 859 the subspaces covering the non-affine supports can be infinite. I think the authors probably mean "countable" in Definition C.8, so that the preimage are made of atoms rather than continuous regions._
>
> Thank you for this question. In fact, in Definition C.8 we want preimage to be finite. Definition C.8 asks for 2 conditions: 1) finite preimage 2) locally around every point in the preimage f is defined by an affine function. Condition 1) is satisfied by the assumption that f is weakly injective. And L859 is dealing with condition 2).
>
> > _8. Throughout the development of the proofs, x can appear as the input or output of the decoder function f, which can be quite confusing and takes time for the reader to follow._
>
> We apologize for this choice of notation and will address this in the final version.
>
> > _9. equation (11), I think the notation p| deserves a specific definition._
>
> Thank you for the suggestion, we will add the corresponding definition.
>
> > _10. Various symbols on page 20 can be much better illustrated by an example of f:R->R that is piecewise affine, where x_0 is a value on the vertical axis, then y's on the horizontal axis, B is a region around one y_i, etc. The concept here is actually quite simple, though notationally heavy, for a reader like me._
>
> Thank you for this excellent suggestion. We believe you are referring to the discussion at L868-870, and we agree this could be developed at a more intuitive level with a figure. We will add this figure along with a discussion in the proof on p. 20.
>
> We are happy to incorporate the other suggestions in the final version of the paper and welcome additional feedback.

---

> > ### Comment · Reviewer_dpBS · 2022-08-08
> > **Thanks**
> >
> > I read the response and maintain my score. I congratulate the authors again on their valuable contribution.

---

### Meta-Review · Area_Chair_TgVM · 2022-08-26

**Recommendation:** Accept
**Confidence:** Certain

**Metareview:**

All three reviewers gave solid recommendations for acceptance. Authors provided a fairly detailed response to each review; each reviewer confirmed they would maintain their positive scores. Clear accept.

**Award:**

No

---

### Decision · Program_Chairs · 2022-09-14

Accept